# Sonographic Diagnosis and Follow-Up of a Rare Large Pre-Patellar Morel-Lavallée Lesion

**DOI:** 10.3390/diagnostics15070883

**Published:** 2025-04-01

**Authors:** Peter Kam-To Siu, Wei-Ting Wu, Levent Özçakar, Ke-Vin Chang

**Affiliations:** 1Department of Orthopaedics and Traumatology, Queen Mary Hospital, Hong Kong; hkpetersiu@gmail.com; 2Department of Orthopaedics and Traumatology, The University of Hong Kong, Hong Kong; 3Department of Physical Medicine and Rehabilitation, National Taiwan University Hospital, Bei-Hu Branch, No. 87, Nei-Jiang Rd., Wan-Hwa District, Taipei 10845, Taiwan; wwtaustin@yahoo.com.tw; 4Department of Physical Medicine and Rehabilitation, National Taiwan University College of Medicine, Taipei 10006, Taiwan; 5Department of Physical and Rehabilitation Medicine, Hacettepe University Medical School, Ankara 06100, Turkey; lozcakar@yahoo.com; 6Center for Regional Anesthesia and Pain Medicine, Wang-Fang Hospital, Taipei Medical University, Taipei 110301, Taiwan

**Keywords:** knee pain, hematoma, degloving injury, ultrasonography, rehabilitation, Morel Lavallee

## Abstract

Morel-Lavallée lesions (MLLs) in the knee are rare. This article presents one of the largest documented cases (13.46 × 12.73 × 3.03 cm), successfully managed non-operatively with clinical and sonographic follow-up. The patient was a 19-year-old male who sustained a bike accident, presenting with immediate gross swelling over the anterior knee, along with bruising and skin abrasions. He was unable to walk and was admitted to the orthopedic ward. A bedside handheld ultrasound examination revealed a large pre-patellar MLL, primarily filled with blood clots. Ultrasound-guided aspiration resulted in a dry tap. The diagnosis was confirmed through magnetic resonance imaging, and conservative management was implemented. At four weeks post-injury, the patient showed significant improvement. He achieved full range of knee motion seven weeks post-injury. Ultrasound imaging revealed a significantly reduced lesion size, with a sub-centimeter thickness. This case highlights the pivotal role of portable ultrasound in the acute diagnosis and follow-up of MLLs. It also demonstrates that, even with a lesion of substantial size, non-operative treatment can be highly effective. These findings provide valuable insights for developing treatment protocols for this rare condition.

**Figure 1 diagnostics-15-00883-f001:**
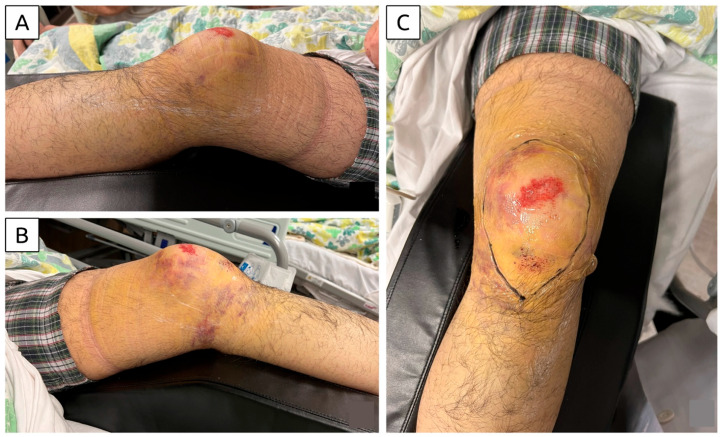
Clinical photographs of the knee after hospitalization. (**A**) Lateral view of the knee demonstrating swelling; (**B**) medial view of the knee showing extensive bruising extending to the dependent region; (**C**) anterior view of the knee highlighting bruising and a superficial abrasion. The extent of the subcutaneous hematoma was outlined on the skin. The photographs belong to a 19-year-old boy—with no significant past medical history—who sustained injuries in a bike accident. He fell forward from the bike, landing on his left anterior knee against a rough ground and sliding forward due to momentum. He reported immediate, gross knee swelling. The examination revealed skin abrasions and bruises but no open wound. Unable to bear weight, he was transported via ambulance and admitted to the orthopedic ward. Clinical assessment identified firm and tender swelling over the anteromedial knee, accompanied by a superficial skin abrasion and diffuse bruising extending over the anterior and medial aspects of the knee. The bruising tracked to the posterior knee in the dependent region. Passive knee range of motion was from 0 to 90 degrees, with tenderness at the end range. Further examination was limited by knee pain. Radiographic evaluation revealed no fractures (see Appendix A).

**Figure 2 diagnostics-15-00883-f002:**
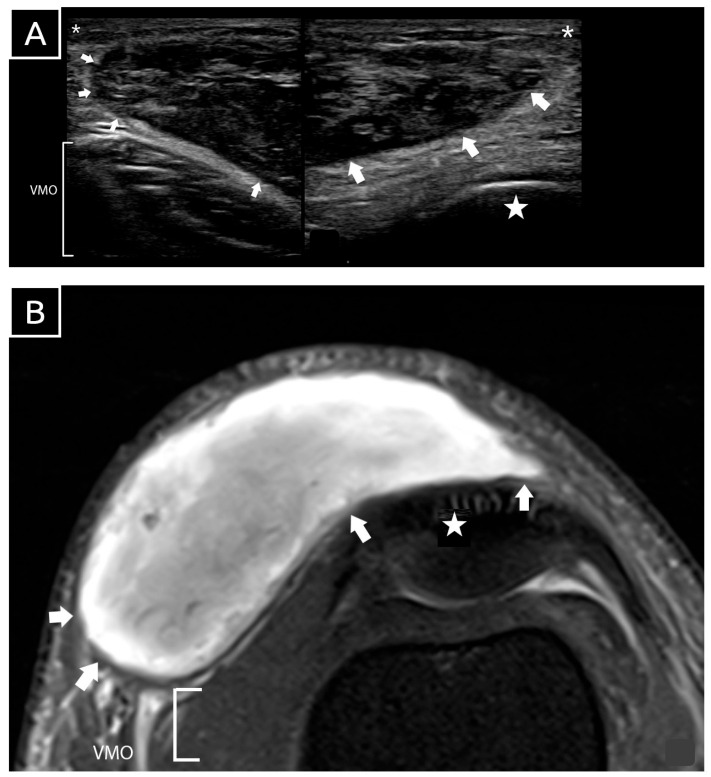
Sonographic image (**A**) and axial magnetic resonance imaging (MRI) (**B**) of the patient’s knee after injury. The location, extent, and contents of the MLL (white arrow) were readily visualized on ultrasound. The sonographic image (A) reveals heterogeneous echogenicity suggestive of blood clots, with no fluid component identified within the lesion. Key anatomical landmarks are labeled: VMO (vastus medialis oblique), asterisk (*) denoting the skin and subcutaneous layer, star (★) representing the bony patella, and white arrow outlining the lesion. Image A was obtained via bedside ultrasound using a Philips Lumify with an L12-4 MHz linear probe, operating under factory default settings for frequency, dynamic range, and time gain compensation. The scan demonstrated a large, well-encapsulated hematoma located in the subcutaneous plane, superficial to the bony patella and exerting mass effect on vastus medialis. The lesion measured 13 cm craniocaudally, 10 cm mediolaterally, and had a maximum thickness of 1.33 cm. The internal contents exhibited heterogeneous echogenicity consistent with blood clots, and no fluid component was observed. Power and standard Doppler imaging detected no vascular flow within or around the hematoma, ruling out active bleeding. The vastus medialis appeared normal, with intact echogenic fibers and fascial covering, showing no signs of injury. The clinical and imaging findings were consistent with a diagnosis of an MLL of the knee. The patient remained hemodynamically stable during hospitalization, with normal laboratory values, including hemoglobin, platelet count, and clotting profiles. An ultrasound-guided aspiration was performed under aseptic conditions using a wide-bore (19 G) needle to target multiple areas of the lesion. No fluid was aspirated, as the lesion comprised primarily blood clots. The patient was managed conservatively with wound dressing, compressive bandaging, ice therapy, non-steroidal anti-inflammatory drugs, and a hinged knee brace locked at 0 degrees to protect soft tissues and minimize shear stress on the hematoma. The patient was allowed full weight-bearing walking exercises while wearing the brace, but excessive walking was restricted. Outpatient follow-up included daily dressing changes and renewal of the compressive bandage to maintain effective compression. MLLs are commonly associated with high-energy trauma and frequently occur alongside fractures of the proximal femur, pelvis, and acetabulum. These traumatic events can lead to the separation of the skin and subcutaneous tissues from the underlying fascia, resulting in the formation of these lesions. Letournel and Judet [1] emphasize the incidence of MLLs in acetabular fractures, indicating that approximately 8.3% of patients with acetabular fractures also experience these lesions. This highlights the importance of recognizing MLLs in the context of severe orthopedic trauma. A systematic review by Shen et al. [2] found that males are roughly twice as likely to experience these lesions compared to females, reflecting the higher incidence of polytrauma among men. However, MLLs are often undiagnosed or diagnosed late, suggesting that their true prevalence may be significantly underestimated.

**Figure 3 diagnostics-15-00883-f003:**
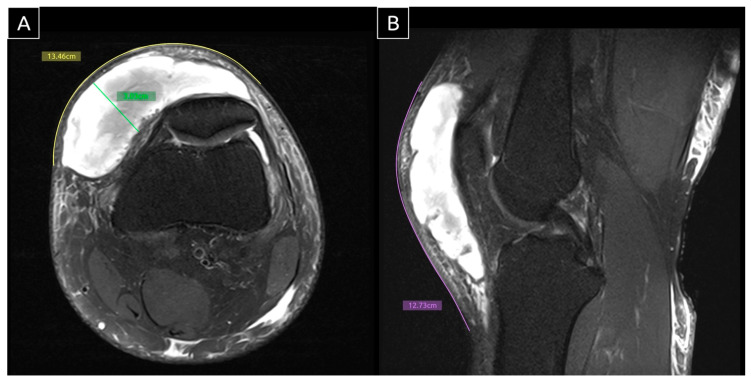
Axial (**A**) and sagittal (**B**) MRI. MRI was performed 11 days after the initial ultrasound scan, with measurements obtained using Horos software for skin-level tracing. The findings were consistent with the prior ultrasound, revealing a large MLL in the knee. The lesion measured 12.73 cm craniocaudally, 13.46 cm mediolaterally, and 3.00 cm in depth at its thickest region—compatible with a Type II MLL based on a history of trauma, characteristic clinical signs, and imaging evidence [3]. The MLL is a closed degloving injury resulting from shearing forces on soft tissues. These forces cause differential tissue sliding and rupture of perforating vessels, leading to bleeding and the formation of a fluid-filled cavity [4]. This cavity is typically filled with blood, lymph, and necrotic fat. If untreated, the inflammatory reaction leads to the formation of a pseudo-capsule, which prevents fluid reabsorption and results in persistent fluid accumulation [5]. This process can result in a chronic MLL, which may persist for months or even years.

**Figure 4 diagnostics-15-00883-f004:**
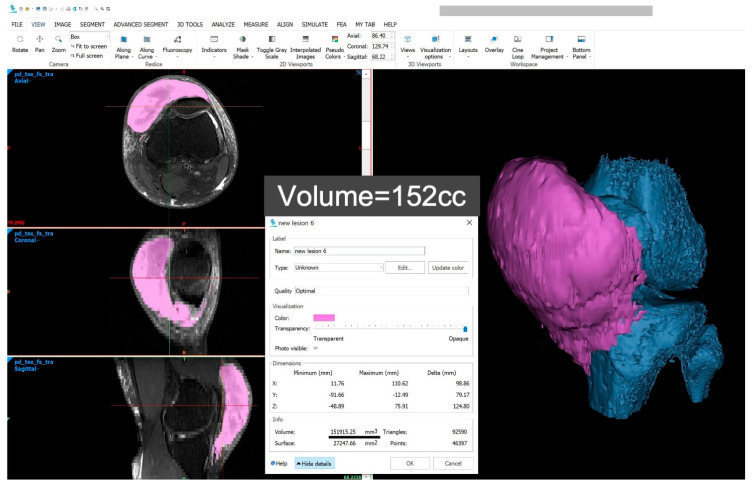
Accurate calculation of lesion volume was achieved using 3D modelling. MRI DICOM data were imported into Materialise Mimics for analysis. Segmentation of the lesion and bones was performed, and a 3D model was generated. MLL is denoted in purple and bony structures of the knee in blue. This allowed precise volume calculation of the lesion, as demonstrated. Vanhegan et al. [6] reviewed over 200 cases of MLLs reported in the literature and documented the sites of occurrence as follows: greater trochanter/hip (30.4%), thigh (20.1%), pelvis (18.6%), knee (15.7%), gluteal region (6.4%), lumbosacral area (3.4%), abdominal area (1.4%), calf/lower leg (1.5%), and head (0.5%). Borrero et al. [4] described MRI findings of four cases of pre-patellar MLLs—with the largest lesion measuring 13.6 × 6 cm and successfully managed non-operatively (with limited reported clinical data).

**Figure 5 diagnostics-15-00883-f005:**
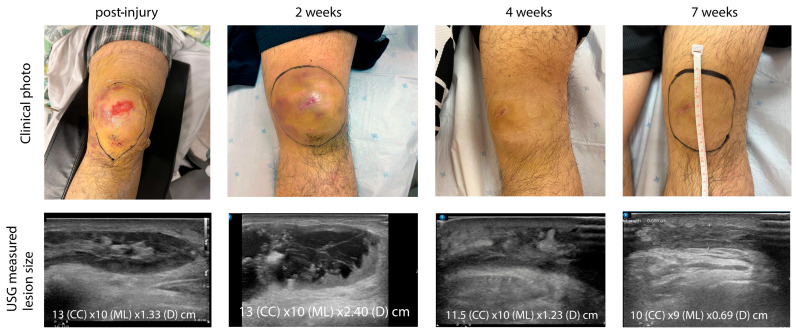
Summary of clinical and sonographic findings at various time intervals. Sonographic images were obtained at the site of the thickest portion of the lesion. The lesion’s border was marked on the skin under sonographic guidance, and corresponding measurements were taken on the skin using a tape measure. Measurements included craniocaudal, mediolateral, and depth dimensions. At two weeks, the patient’s knee pain had improved significantly, and he was able to walk unaided. The skin abrasion was healing, and no skin necrosis was observed. The active knee range of motion was from 0 to 70 degrees. Ultrasound imaging revealed that the lesion measured 2.4 cm in thickness and was composed of an organized blood clot without a fluid component. Daily dressing and bandaging were continued, and the hinge knee brace was adjusted to allow a range from 0 to 90 degrees. At four weeks post-injury, the knee swelling had resolved significantly. There was no tenderness, and the skin abrasion had fully healed. The patient achieved a full range of knee motion. Ultrasound imaging demonstrated a reduction in lesion size. Assuming the hematoma had an oval shape, its volume decreased from 181.05 mL to 65.02 mL, representing a 64% reduction. The knee brace was removed, and dressing and bandaging were discontinued.

**Figure 6 diagnostics-15-00883-f006:**
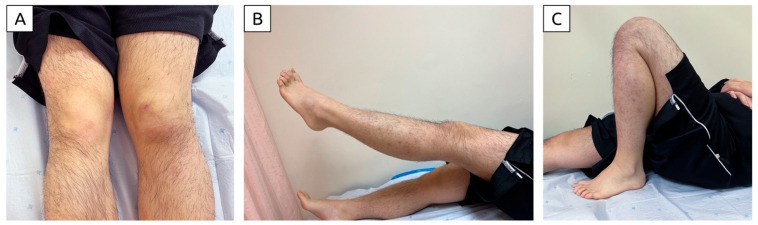
Photograph taken seven weeks post-injury demonstrates (**A**) mild residual swelling compared to the unaffected knee, (**B**) full knee extension, and (**C**) full knee flexion. The patient was referred for quadriceps strengthening and progressive functional training. This article highlights the pivotal role of portable ultrasound in managing MLLs of the knee. It is convenient, cost-effective and can be performed promptly/safely in acute clinical settings, eliminating the need for patient transfer. Doppler ultrasound enables the assessment of active blood flow, aiding in the identification of active bleeding. Ultrasound also facilitates guided interventions, ranging from fluid aspiration in acute lesions to sclerosant injection in chronic cases. Its ability to be repeated at frequent intervals allows for close monitoring of lesion progression and treatment response. If there is any sign of progression despite initial non-operative treatment (especially for large lesions, like in this case), early surgical treatment should be planned. Surgery plays a major part in the overall treatment options for large MLLs of the knee. The risks of leaving a large MLL untreated include infection and pseudo-capsule formation, both of which are difficult to manage. Thus, ultrasound serves as a very useful tool in addition to clinical assessment to alert clinicians of the need for early surgery. The literature also suggests that MLLs are likely underreported, as smaller knee lesions may be overlooked or underestimated during initial evaluation. Without timely and appropriate treatment, these lesions can progress to a stage requiring surgical intervention. Functional morbidities associated with MLLs include knee flexion stiffness and quadriceps weakness, often resulting from prolonged immobilization. Conversely, overly aggressive rehabilitation may disturb hematomas and increase the risk of skin necrosis. Frequent clinical and sonographic assessments provide the foundation for safe, individualized rehabilitation protocols. Finally, sonographic evaluations enhance communication with patients, aligning treatment goals to ensure patient-centered care [7,8]. This case highlights the crucial role of portable ultrasound in the acute diagnosis and follow-up of MLLs. It also demonstrates that non-operative treatment can be highly effective, even for a lesion of substantial size. These findings offer valuable insights for developing treatment protocols for this rare condition. Finally, the findings from a single case may not be generalizable, and further studies are needed to validate these observations.

## Data Availability

All the ultrasound and 3D modelling images were produced by the first author of this article, P.K.-T.S. They are available upon reasonable request to him.

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
