# Peer review of "Sonographic Diagnosis and Follow-Up of a Rare Large Pre-Patellar Morel-Lavallée Lesion"

_diagnostics, 2025, doi:10.3390/diagnostics15070883_

Round 1

Reviewer 1 Report

Comments and Suggestions for Authors

The abstract should include a brief mention of the key outcomes and specific measurements of the lesion after treatment

Include about 100 more words on the epidemiology of MorelLavallee lesions in non-sporting environments

what were the exact ultrasound settings used to differentiate between types of tissue in the lesion

include a kind of statistical analysis to substantiate the effectiveness of the treatment protocol, even if the study is a case report

there are plenty biases or limitations introduced by the single subject study design

state the implications for changes in clinical practice / future research

Author Response

Dear reviewer:

     Please kindly refer to the attached file for our responses to your suggestion. Thank you for your patience and support!

Reviewer 2 Report

Comments and Suggestions for Authors

Dear authors, I had the opportunity to review your manuscript. I hope that my comments and remarks are helpful and constructive.

The author presented an interesting case; however, I have some concerns regarding the diagnosis and management plan.

-I noticed that none of the authors were radiologists. Was the US diagnosis and follow up performed by an orthopedic surgeon or a physiotherapist? Please clarify. As most surgeons are not well trained to use the US machine with such precision as described in the case report.

-Regarding the diagnosis, the authors mentioned the following:

“The bruising tracked to the posterior knee in the dependent region. Passive knee range of motion was from 0 to 90 degrees, while active motion was limited due to pain. Active quadriceps contraction was preserved without tenderness. Ligamentous testing showed no laxity of the collateral or cruciate ligaments. Radiographic evaluation revealed no fractures, patella alta (which would suggest patellar tendon rupture), or patella baja (which would indicate quadriceps tendon rupture).”

With all due respect, the examination is misleading. The authors reported that the patient could not perform active knee motion due to pain, and at the same time, the authors could elicit all ligamentous injury evaluations and confirm that the ligaments were intact and that the patient did not complain of pain during the examination.

Second, even with a rupture of the patellar tendon, some patients could perform active extension or maintain extension due to the action of the retinaculum.

-Please provide the radiographic images.

-What were the diagnostic criteria of Morel-Lavallee Lesion?

-Clinical images presented in Figure 5 appear as if they were manipulated.

-In Figure 6, what are the reasons for the apparent atrophy of the VMO on the right side?

Comments on the Quality of English Language

Shorten the sentences and perform language editing

Author Response

(The authors gave the same response as above.)

Reviewer 3 Report

Comments and Suggestions for Authors

I would first of all commend the authors on the quality of images used to describe this case. All images from the clinical photographs / ultrasound and especially the 3D modelling are of excellent quality. 

Having managed several MLL around the knee myself, I would recommend that surgery does play a large part in overall treatment options. Generally for lesions > 5 cm, I would proceed with surgical decompression in a semi-elective / planned manner usually at around the 3-6 week stage once the soft tissues are confirmed to have been settled. The risks of leaving them are largely a) infection and b) pseudocapsule formation. The latter can be difficult to manage within the chronic setting. It is entirely acceptable of course to trial a period of conservative treatment beforehand and revert to non-operative treatment if the lesion settles as is shown in this case. Having reviewed the literature I would suggest that above follows the literature, although different authors may undertake slightly different approaches. 

The only modification I would advise therefore is for the authors to mention / highlight the (established) role of surgery should conservative treatment fail. Otherwise I would recommend this manuscript for publication as it does add value. 

Author Response

(The authors gave the same response as above.)

Round 2

Reviewer 2 Report

Comments and Suggestions for Authors

Dear authors, thanks for your responses. Regarding the response to my first comment, I was not asking regarding the authors' expertise; publishing 300 articles on U/S does not guarantee that this person is the best for performing U/S diagnosis. I found all the responses are not satisfactory; for example, the Lachman test does not require active knee motion, but imagine holding the knee in the examiner's hand and pulling the tibia forward with some force to test the stability of the ACL, is this maneuver "nonpainful," I doubt.  All the best to the authors.

Author Response

Dear reviewer:

   Please kindly check our reply. Your patience and kindness are highly appreciated. 

Round 3

Reviewer 2 Report

Comments and Suggestions for Authors

Dear authors, Unfortunately, you corrected the knee examination based on my worries, which raises further concerns. Did you really examine the knee? With all due respect, I am not convinced with your correction. Even more, the corrections made me more suspicious. All the best.